# A Real-Time Computer Vision Based Approach to Detection and Classification of Traffic Incidents

**Mohammed Imran Basheer Ahmed** [1,*], **Rim Zaghdoud** [1], **Mohammed Salih Ahmed** [1], **Razan Sendi** [1], **Sarah Alsharif** [1], **Jomana Alabdulkarim** [1], **Bashayr Adnan Albin Saad** [1], **Reema Alsabt** [1], **Atta Rahman** [2] **and Gomathi Krishnasamy** [3]

1. Department of Computer Engineering, College of Computer Science and Information Technology, Imam Abdulrahman Bin Faisal University, P.O. Box 1982, Dammam 31441, Saudi Arabia
2. Department of Computer Science, College of Computer Science and Information Technology, Imam Abdulrahman Bin Faisal University, P.O. Box 1982, Dammam 31441, Saudi Arabia
3. Department of Computer Information System, College of Computer Science and Information Technology, Imam Abdulrahman Bin Faisal University, P.O. Box 1982, Dammam 31441, Saudi Arabia
* Correspondence: mbahmed@iau.edu.sa

**Abstract:** To constructively ameliorate and enhance traffic safety measures in Saudi Arabia, a prolific number of AI (Artificial Intelligence) traffic surveillance technologies have emerged, including Saher, throughout the past years. However, rapidly detecting a vehicle incident can play a cardinal role in ameliorating the response speed of incident management, which in turn minimizes road injuries that have been induced by the accident's occurrence. To attain a permeating effect in increasing the entailed demand for road traffic security and safety, this paper presents a real-time traffic incident detection and alert system that is based on a computer vision approach. The proposed framework consists of three models, each of which is integrated within a prototype interface to fully visualize the system's overall architecture. To begin, the vehicle detection and tracking model utilized the YOLOv5 object detector with the DeepSORT tracker to detect and track the vehicles' movements by allocating a unique identification number (ID) to each vehicle. This model attained a mean average precision (mAP) of 99.2%. Second, a traffic accident and severity classification model attained a mAP of 83.3% while utilizing the YOLOv5 algorithm to accurately detect and classify an accident's severity level, sending an immediate alert message to the nearest hospital if a severe accident has taken place. Finally, the ResNet152 algorithm was utilized to detect the ignition of a fire following the accident's occurrence; this model achieved an accuracy rate of 98.9%, with an automated alert being sent to the fire station if this perilous event occurred. This study employed an innovative parallel computing technique for reducing the overall complexity and inference time of the AI-based system to run the proposed system in a concurrent and parallel manner.

**Keywords:** accident severity classification; postcollision vehicle fire detection; vehicle detection; DeepSORT tracking; object detection; computer vision

## 1. Introduction

Road traffic accidents (RTAs) are churning the world, as they are one of the most salient and conspicuous public health and development calamities around the world. The World Health Organization's (WHO) Global Status Report on Road Safety affirmed that over the last decade, the annual fatality rate per 100,000 individuals due to RTAs in the Kingdom of Saudi Arabia has substantially escalated from 17.4 to 27.4, which is a dismaying and perturbing situation [1]. Accordingly, the current trends divulge that by 2030, road traffic injuries are predicted to be the seventh leading cause of death across all age groups in Saudi Arabia if no adequate preventive measures are effectuated [2]. In fact, a prominent indicator of the survival rate after an accident is the time frame between the vehicle accident's occurrence and when the emergency medical or fire service

dispatches an ambulance to the traffic accident scene in accordance with the severity of the accident. Providing expeditious and well-timed postaccidental care while utilizing AI-based technologies can play a predominant role in resolving this societal problem by lessening the fatal accidental human damage rate [3].

In accordance with Vision 2030's strategic objective to reduce traffic incidents and minimize their ruinous repercussions [4], this proposed work aims to effectively develop a real-time traffic incident detection and alert system while integrating a deep learning and computer vision-based approach to eliminate the delay between the incident's occurrence and the first responder dispatch. The speed at which the medical service arrives to the incident's victim can in fact make a noteworthy difference in changing the impending sequence of events of the victim's life [5]. Hence, the proposed AI system is intended to be displayed as a prototype interface to demonstrate the effectiveness of capturing the occurrence of a vehicle accident and accurately pinpointing the accident's level of solemnity and seriousness. Accordingly, the exact time and location of the accident will be retrieved, and based on the accident's severity level, an immediate alert message that includes the elicited information will be promptly sent to the nearest medical center if needed to productively undertake felicitous and pertinent measures.

Additionally, in certain occurrences, a fire may take place during the accident's occurrence. Postaccident vehicular fires pose a perilous and unpredictable risk to the vehicle passengers involved in the incident. After a collision has taken place, an automobile can in fact ignite when it overheats due to technical defects, its malfunctioning and inoperative fuel system leaks, or when its engine combusts [6]. The severe consequences of such a series of events poses a predominant concern to the environment and public as large due to the inhalation of burning toxic fumes and due to the burning injuries or deaths that could result from such incidents [6]. Subsequently, to ensure that an immediate response will be initiated, a postcollision vehicular incident detection model is leveraged within the proposed work to overall mitigate and prevent such consequential threats from occurring in the Kingdom. When such an incident has taken place, an instant and automatic email alarm will be straightforwardly sent to the fire station to clear the incident as soon as possible with limited delay before causing a catastrophic cycle of undesirable events.

As a matter of fact, a multitudinous number of measures have been undertaken by the Ministry of the Interior (MOI) to ameliorate traffic safety by promoting road monitoring technologies including an automated traffic management system, known as Saher, that has been leveraged in a 360-degree traffic camera to constructively detect road traffic violations in order to increase traffic safety in the KSA [7]. However, the Kingdom has not yet tackled the societal problem of the detection of traffic incidents. Scientific researchers have, in fact, turned their curiosity and attention towards utilizing several compelling and engrossing computer vision and deep learning-based techniques to automatically detect the occurrence of a traffic accident within its immediate stages to send an alert to the entailed authority. Promising results using the most customarily utilized deep learning techniques, including the CNN, YOLO, Faster-RCNN, and D-CNN algorithms have been selected [8–10]; however, no previous works have aimed towards autonomously classifying an accident's severity level and detecting postcollision vehicle fires. Consequently, to pictorially envision how the proposed vehicle incident detection and alert system will be executed in the hardware, a prototype graphical user interface (GUI) using the QT Designer software development kit is employed as this will lead towards assessing the system's performance by envisaging the system's overall functionalities and capabilities. In the KSA region, traffic accidents are mainly evident on the national highways due to several reasons, such as sandstorms, camels crossing, etc. Due to huge geographical road expansions, physical check posts and patrolling are not applicable. Hence, there is a dire need for such systems to get timely alerts and responses. Currently, no study has been conducted in this regard. A state-of-the-art dataset from the Middle Eastern region has been investigated in this regard.

The key contributions of the study are as follows:

1. A comprehensive system model to detect:

    a. road accidents,
    b. their severity, and
    c. postaccident fires.

2. Investigation of AI and computer vision-based approaches for object and event detection.
3. Testing and validating of the proposed model by contrasting with the state-of-the-art techniques.
4. Detection, alert, and postaccident emergency response mechanisms.

The rest of the paper will be structured as follows. Section 2 will provide a comprehensive overview of the previous works that are related to the proposed study. Section 3 extensively discusses the technical aspects of the three computer vision models and further elucidates each model's overall methodological architecture that was designed to constructively implement the AI-based system. Thereafter, Section 4 provides a statistical analysis of the experimental results, their interpretations, and the experimental conclusions that can be drawn by assessing the differences between the selected algorithms in terms of detecting vehicles, classifying the accidents' severity levels, and detecting postcollision vehicle fires. Discussions of the findings and interpretations, as well as the working hypotheses, are presented in Section 5. Finally, Section 6 concludes the study, provides recommendations based on the results, and suggests future work.

## 2. Related Work

Over recent decades, extensive research has been conducted in the field of intelligent transportation systems that are focused on developing automatic incident detection systems for handling the many day-to-day occurrences within these systems, such as accidents, traffic congestion, and jams. For truly secure smart cities, it remains crucial to attain real-time situational awareness, despite the innovations that have sparked smart city innovation over the past several decades.

In [3], a computationally inexpensive three-stage deep learning-based architecture was proposed to detect car accidents accurately and automatically with minimum hardware requirements. Accordingly, in the first stage, 200,000 raw images are down sampled and Gaussian noise is added to detect vehicles in a moving traffic environment using the Mini-YOLO object detection algorithm [3]. The detected vehicles are then transferred to the vehicle tracking stage to track multiple vehicle objects in the video frame and keep track of each vehicle's damage status in case an accident occurs. The final stage is the classification stage, where the authors in [3] trained the RF, CNN, and SVM algorithms to effectively classify the vehicle images into either the damaged or the undamaged classes. The experimental results that were retrieved from [3] state that in the tracking stage, the Mini-YOLO model attained an AP score of 34.2 and a runtime of 28 frames per second. Additionally, in the classification stage, the SVM with the radial basis function kernel attained a precision score of 96%, a recall of 94%, and an AUC score of 96% [3].

In [8], an object detection algorithm known as YOLOv3 was utilized to detect abnormal situations on the roads and to sufficiently avert secondary accidents. A real-time notification application was constructed by implementing AI CCTV. Accordingly, the FFmpeg software was productively leveraged to capture 700 frames of vehicle accidents from a series of vehicle collision videos to construct the dataset. A rotation of 90 and 180 degrees was performed on the image dataset to relatively increase its size to 2000 images. Thus, the custom weights of the proposed YOLOv3 model obtained a mean average precision of 82.36% and an intersection over union threshold of approximately 50% [8]. The finalized deep learning model was embedded into Django and Flask servers, and a warning alert was then sent via a Firebase Cloud Messaging (FCM) platform upon the occurrence of an accident or collision.

Likewise, researchers in [9,10] also aimed to present an efficacious solution to lessen and reduce the overall road accident rate on highways by conducting a deep learning-based accident detection system. To effectuate the system, CCTV cameras were mounted on highways. In [9], convolutional neural networks (CNNs), which work upon the ReLU and Sigmoid activation layers and a loss function to eliminate any noise that may have gathered within the road accident video frames, were used, whereas in [10], a DL model that combined CNN (inception v3) and LSTM was leveraged to classify whether or not an accident had occurred in the video frame. The model in [10] was then implemented on a Raspberry Pi using Keras, TensorFlow, and OpenCV. Furthermore, in [10], a GSM module was obtained to create an alarm system to send an SMS to the nearest hospital or police station if the prediction exceeded a threshold of 60%, whereas in [9], an email alert was used instead of an SMS message. The alert message in both [9,10] contained information about the accident's time occurrence, the location of the accident, and the frame for further analyses. Both [9,10] utilized a CNN-based architecture, hence, the proposed experimental work in [9] attained an accuracy of 93%, and in [10], an accuracy averaging 92.38% was achieved. In [11], the paper introduced a supervised deep learning framework solution to establish a car crash detection system, which can function smoothly and transmit critical information to the appropriate authorities without any delay. The dataset used consisted of CCTV video clips from YouTube of different car crash conditions from different Middle Eastern regions. The car crash detection system was divided into three phases: vehicle detection, vehicle tracking and feature extraction, and accident detection. In the first phase, the Mask RCNN (Region-based Convolutional Neural Networks) was used to segment and build pixel-by-pixel masks for each item in the video. The Centroid Tracking algorithm was then used to effectively track the vehicle to observe the cause of the accident, which was then classified as being due to speed acceleration, trajectory anomaly, or change in angle anomaly. A 71% detection rate and 0.53% false alarm rate using the accident videos were successfully obtained under different surrounding environmental conditions [11]. Various intelligent methods were applied in the field of medicine and other vital fields as detailed in the research work [12–16].

## 3. Methodology

The proposed neoteric and computer vision-based system imparts a five-phase modernized solution to detect traffic incidents in both sparse and dense traffic flow environments while considering low-visibility and differing environmental weather conditions. Accordingly, Figure 1 illustratively depicts the top-tier methodological workflow of the proposed vehicular traffic incident detection and alert system.

The first phase as shown in Figure 1 is the vehicle detection and tracking phase. It is accountable for processing a real-time CCTV traffic surveillance video stream to detect all types of vehicles, including trucks, cars, buses, and motorcycles, in a moving traffic environment. To refine the overall robustness of the model, the DeepSORT tracking algorithm was integrated within the system's architectural framework to accurately track the vehicles by returning an unrepeatable identification number for each individual vehicle that appears in each incessant stream of video frames. Incorporating such object-tracking aids determination of the overall status of all vehicles in case an accident takes place. The second phase detects the occurrence of an accident and classifies its corresponding solemnity and severity. Once either a moderate or severe accident has taken place, then all the upcoming video frames are sent to the postcollision vehicle fire detection model, which is the third phase of the proposed work. This model will detect the occurrence of a fire after an accident has taken place. Subsequently, the execution of the email alert system is the fourth phase, as its pre-eminent intent is to effectively provide the competent authorities with entailed and integral information to grant them the capability to take actions that can provide protection and safety for the public. Thus, they would be more likely to take protective action in eliminating the delay between the incident's occurrence and the first responder's dispatch. Once the three models along with the alert system are

constructed, the final model integration phase is implemented to embed the three premier object detection and image classification models into a single unified and automated system to be able to undergo the alarm aspect of the proposed system's pipeline.

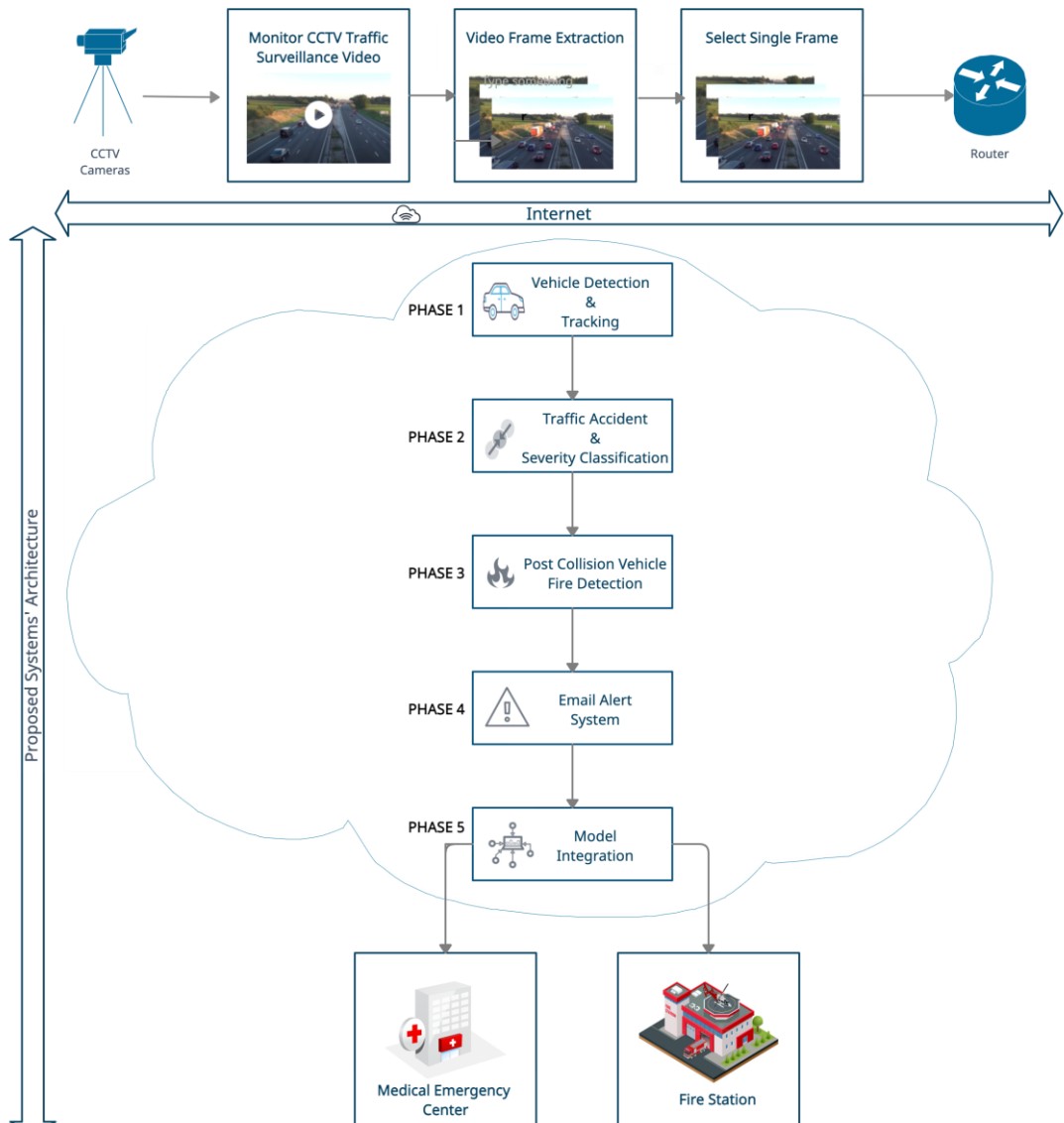

**Figure 1.** Methodological workflow of the proposed system.

### 3.1. Phase 1: Vehicle Detection and Tracking

A premier factor of the real-time traffic incident detection and alert system is to constructively detect and track vehicles in a traffic environment to effectively monitor the vehicles for any unforeseeable and incalculable sequence of events. Accordingly, the first phase's notable objective is to filter out all the objects present in each forthcoming sequence of traffic video frames to only preserve correctly detected and recognized vehicle objects. Three versions of a convolutional neural network (CNN) known as YOLO (You Only Look Once), including the YOLOR, YOLOv5, and YOLOv4-Tiny, have been utilized to undergo the vehicle detection aspect of the AI-based system as it is one of the most popular computer vision models for real-time object detection due to its astonishing speed and efficiency [17].

Once the vehicles have been detected in each of the extracted traffic surveillance video frames while leveraging the YOLO object detector, the next task of the proposed model is to validly keep track of each of the detected vehicular objects. This task is implemented

by integrating a customarily utilized and elegant tracking algorithm. That algorithm is an extension of the Simple Online and Realtime Tracking (SORT) algorithm known as the DeepSORT tracker. Accordingly, the tracker is adopted within the architecture to overcome a series of unwanted factors that are caused by differing camera motions during real-time, CCTV-based video detection. Consequently, the detection aspect of the framework is accountable for detecting the vehicle object that appears within a single frame, whereas the tracking approach contains vehicle objects that are currently being monitored by assigning a unique and distinctive ID by allocating a bounding box for each individual tracking element that contains the objects' associated IDs. The overall computer vision-based framework for the vehicle detection and tracking model is displayed in Figure 2.

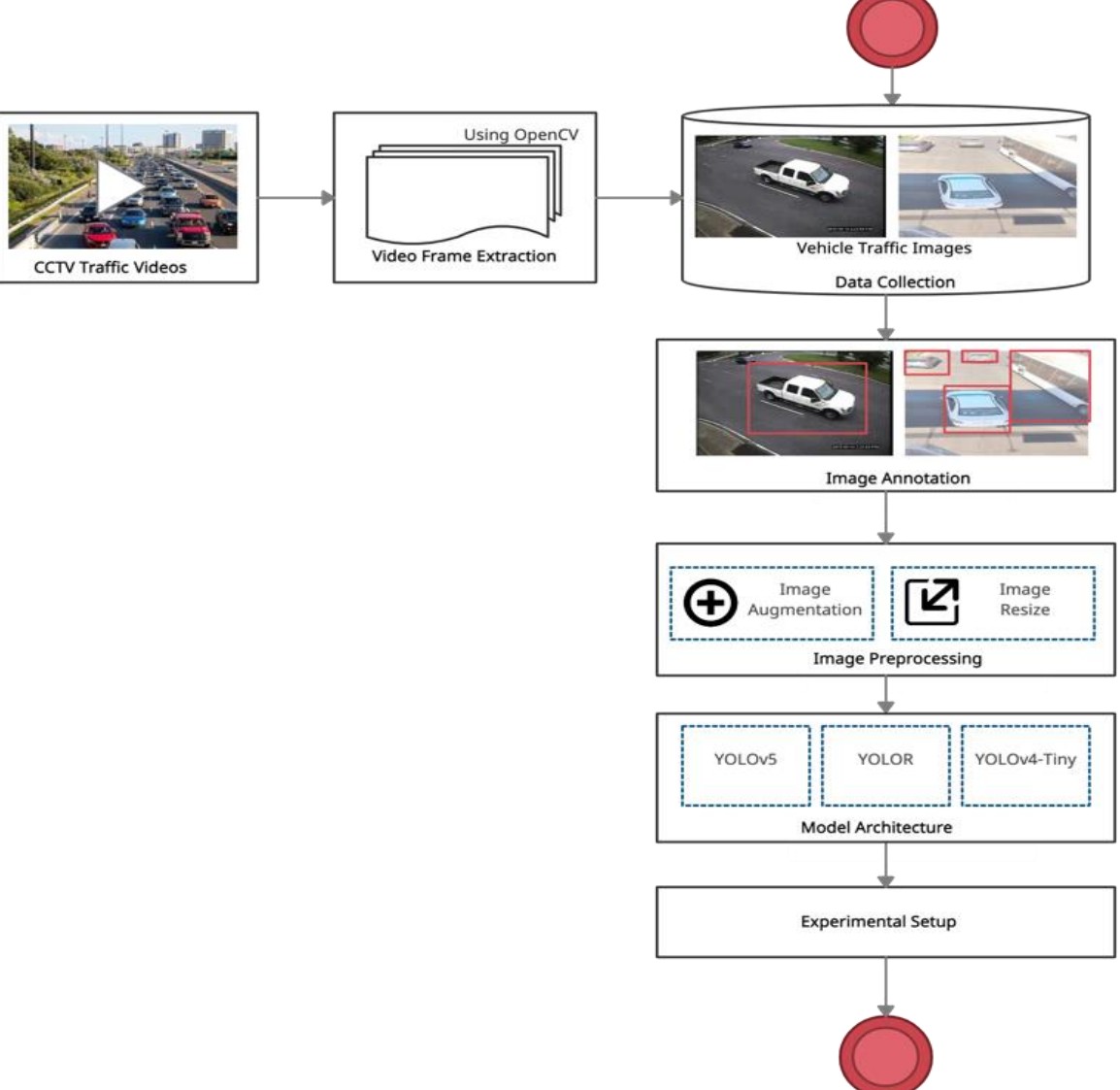

**Figure 2.** Phase 1 Flowchart for the Detection and Tracking of Vehicles.

### 3.1.1. Data Collection

Pinpointing vehicles in real time is a challenging task due to the multitude of road scenes, differing viewpoint variations, manifold illumination levels, and weather conditions. Accordingly, a multitudinous number of large scale open-source datasets that are publicly available are specifically annotated to detect vehicles, but such datasets only tackle high-quality vehicle images which remains a crucial bottleneck for the detection of vehicles in CCTV video streams. To alleviate and attenuate the bottleneck, a custom vehicle

detection image dataset composed of 8000 annotated images containing approximately 36,400 vehicle instances extracted from CCTV traffic surveillance footage was utilized. The datasets for vehicle, accidents, and fires were obtained from various open sources, such as [18–20], and subsequently preprocessed and annotated. A sample of some of the divergent circumambient conditions that have been taken into consideration during the dataset collection phase are illustratively shown in Figure 3.

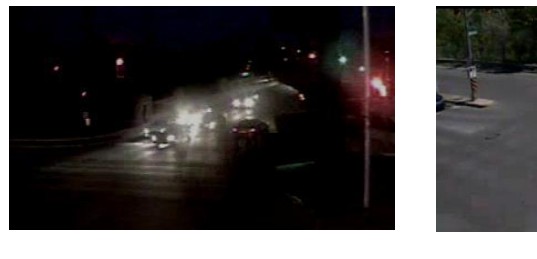 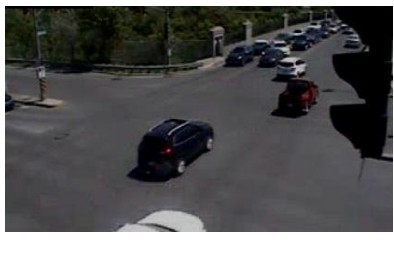 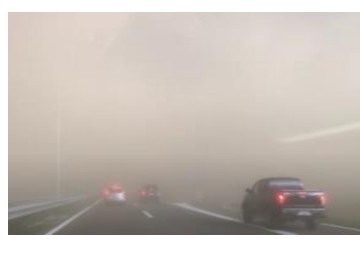

(**a**)                               (**b**)                               (**c**)

**Figure 3.** Image Sampled from the Vehicle Dataset: (**a**) Nighttime Images, (**b**) Daylight Images, (**c**) Sandstorm Images.

### 3.1.2. Image Preprocessing

Image preprocessing is considered to be the foundation of the proposed deep learning model, as it plays a crucial role in transforming the raw data into a suitable and comprehensible format [21]. Moreover, the preprocessing step aids in reducing training time, increasing the model's accuracy, and improving the model's overall efficiency by removing noise and unnecessary data from the model. The first and most overriding preprocessing step in this dataset is to resize the images. This step is specifically undertaken because deep learning algorithms tend to be trained more quickly with smaller images, whereas larger images require more learning and storage consumption, thus leading to a more complex computation. As a result, the three YOLO algorithms were resized to the standard resolution size of $416 \times 416$, as the YOLO algorithm must attain a pixel resolution that is a multiple of 32.

Consequently, in deep learning, data augmentation plays an essential role in improving the model's evaluation outcomes and maximizing the benefits of the dataset. Through image augmentation, such intelligent neural network architectures can become more robust by creating differing variations in the images that reflect real-world scenarios. Thus, applying such augmentations to the image dataset leads towards avoiding the expense of collecting and labeling additional CCTV-based vehicle images and videos. The brightening and darkening augmentation method was set between $-25\%$ and $+25\%$ to make the model more robust and adaptable to abrupt changes in radiance. The Gaussian blur transformation was also used with a kernel size of 1.75 pixels to specify the amount of blurring that will be applied to the dataset.

### 3.1.3. Experimental Setup

Before undergoing the training procedure for the YOLOv5, YOLOv4-Tiny, and YOLOR algorithms, the image-based vehicle dataset was split using the 80/10/10 splitting criterion in which approximately 80% of the collected and annotated images were utilized for effectively training the dataset, and the remaining 20% was evenly and uniformly split between the validation and testing datasets. Furthermore, a vehicle tracking algorithm known as DeepSORT was integrated within the YOLO architecture, as this framework is designed to assign each vehicle with a distinctive ID number to track the targeted objects' movements and overall motion in a real-life setting. The DeepSORT tracker was specifically selected because it offers an improved approach that uses another distance metric based on the appearance of the object. The approach involves building and constructing a classifier over the custom dataset, training it up until it has achieved a good level of accuracy, mAP,

and recall, and then stripping out the final classification layer. Classical architecture leaves a dense layer for producing a single feature vector which may later be classified. This feature vector then becomes the "appearance descriptor" of the object, namely, the vehicle. After embedding this tracker within the algorithms, the three object detectors were then trained, validated, and tested on a combination of traffic surveillance images and videos that were comprised of manifold illumination levels and weather conditions to determine and retrieve the algorithm that could explicitly label each image according to its acquired class label. The damage status of a detected vehicle can then be tracked by the constructive implementation of this model.

*3.2. Phase 1: Traffic Accident and Severity Classification*

Assessing the traffic accident's level of damage by automatically classifying its severity level into either moderate or severe is an extremely crucial factor of the proposed computer vision approach, as an email alert will be sent based on the accident's severity level. Through the utilization of this approach, medical assistance will arrive to the entailed and highly urgent traffic accident scene where the vehicle passenger will require immediate assistance in a much faster manner without losing any valuable time by immediately notifying the competent authorities without holding back once a severe accident has taken place.

Accordingly, to tackle and construct an efficient traffic accident and severity classification model in a real-time manner, the sections shown in Figure 4 procedurally elucidate the methodological process that was constructively designed to detect and classify an accident's severity level, starting from the data collection stage up until the end of constructing the model's experimental setup. At this point, multiple object detection algorithms, including the YOLOv5, YOLOR, and Faster R-CNN algorithms, were tested for this model while considering the high necessity for speed, accuracy, and recall.

3.2.1. Data Collection

A custom dataset, comprised of a combination of collected images from online sources and extracted frames from CCTV traffic accident compilation videos from YouTube, has been created. However, the collected and retrieved images and extracted frames lacked in terms of quality and resolution, and thus the model would suffer by being impotent and incapacitated in retrieving integral features that would prominently aid towards differentiating between moderate and severe accidents [22]. Consequently, semirealistic video game videos that portrayed differing vehicle incidents were amalgamated within the current set of gathered images to enhance the model's overall capability. Accordingly, using an open-source image annotating tool, 5185 traffic accident image frames were manually annotated based on the accident's severity level. The targeted accident object was labelled as either a moderate accident or a severe accident.

3.2.2. Image Preprocessing

As most of the images in the database were directly scraped from the web, a large portion, particularly the semirealistic accident images, consisted of high-quality images. To increase the generalization of the model in deployment, the brightness augmentation technique was added to mimic inputs taken directly from low-resolution CCTV cameras. This procedure caused a significant increase in the model's performance at inference time. Additionally, all images and extracted video frames within the accident severity dataset were resized to a fixed width and height of $416 \times 416$ as part of the model's preprocessing phase, as downscaling leads towards the establishment of an optimized model that aims towards simultaneously diminishing the model's complexity and memory consumption issues.

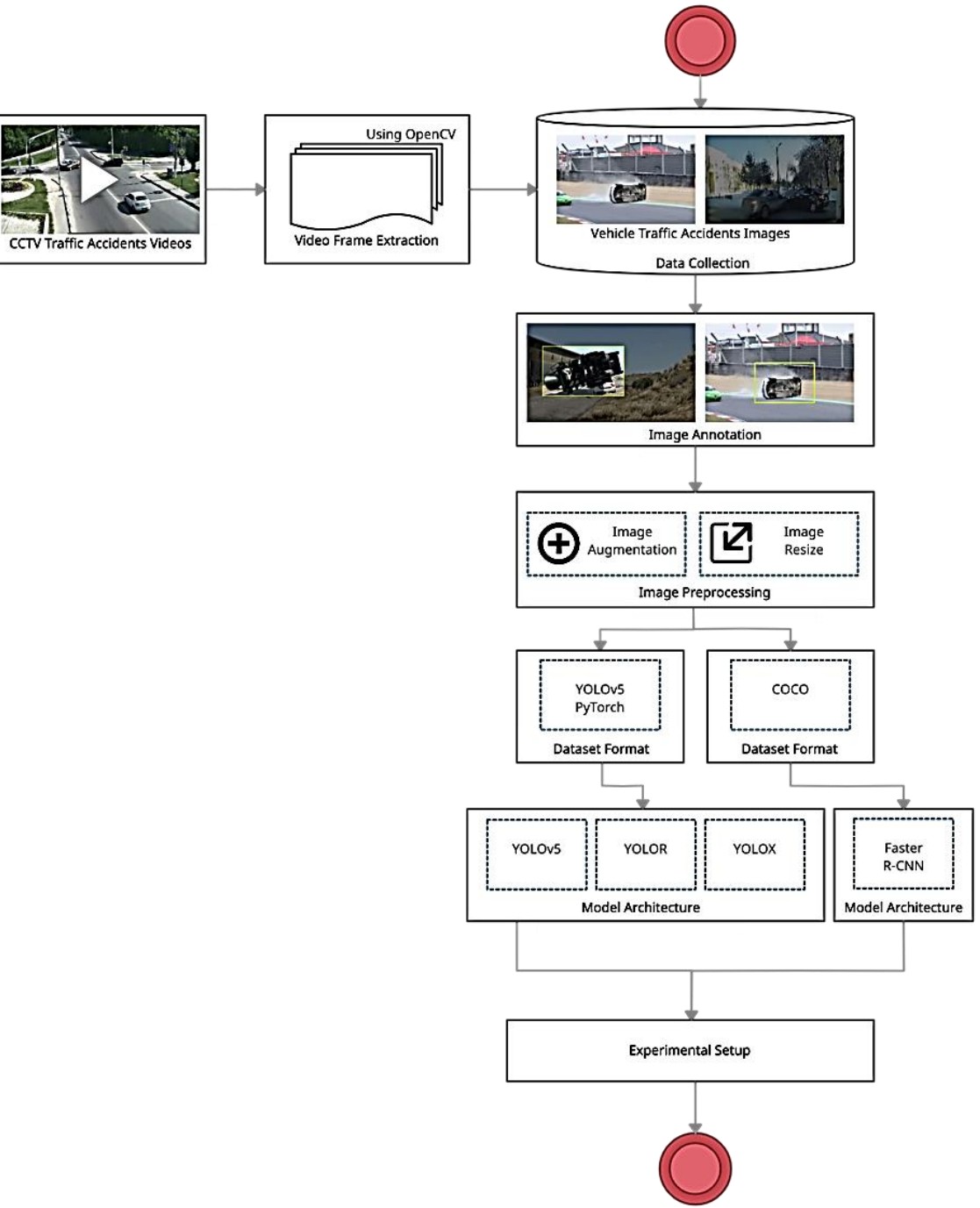

**Figure 4.** Phase 2 Flowchart for the Classification of Accident Severity Level.

3.2.3. Experimental Setup

To effectively construct a reliable and computationally inexpensive real-time severity classification model while leveraging the annotated and preprocessed dataset that was assembled in Section 3.2.1, the model underwent the training, validation, and testing stages of the computer vision-based pipeline for the YOLOv5, YOLOR, and Faster R-CNN object detection algorithms, which have been selected based on their excellent capabilities in accurately detecting and correctly classifying the objects' classes. The dataset was first partitioned into three distinctive sets, as 80% of the data was split into the training set, 10% for the validation set, and the remaining 10% was leveraged for testing the model to attain an unbiased viewpoint on the model's overall performance. Adjusting

certain hyperparameter values, including the epoch and batch size hyperparameters, before feeding the four selected CNN-based algorithms was a crucial step that was implemented before feeding the models into the training phase.

### 3.3. Phase 3: Postcollision Vehicle Fire Detection

Postcollision vehicle fires are one of the most perilous and hazardous traffic incidents, as they can cause a catastrophic number of grievous casualties while endangering human lives. Even though such incidents are sparse events, as they are only accountable for a very small percentage of fire events, they tend to pose a double threat to vehicle occupants, and the loss of human lives and severe bodily injuries due to postincident vehicle fires are in fact still a consequential problem that must be immediately resolved. To completely diminish and abolish this threat, a postcollision vehicle fire incident detection model that detects the occurrence of a fire after a collision is embedded within the proposed architectural framework, as shown in Figure 5.

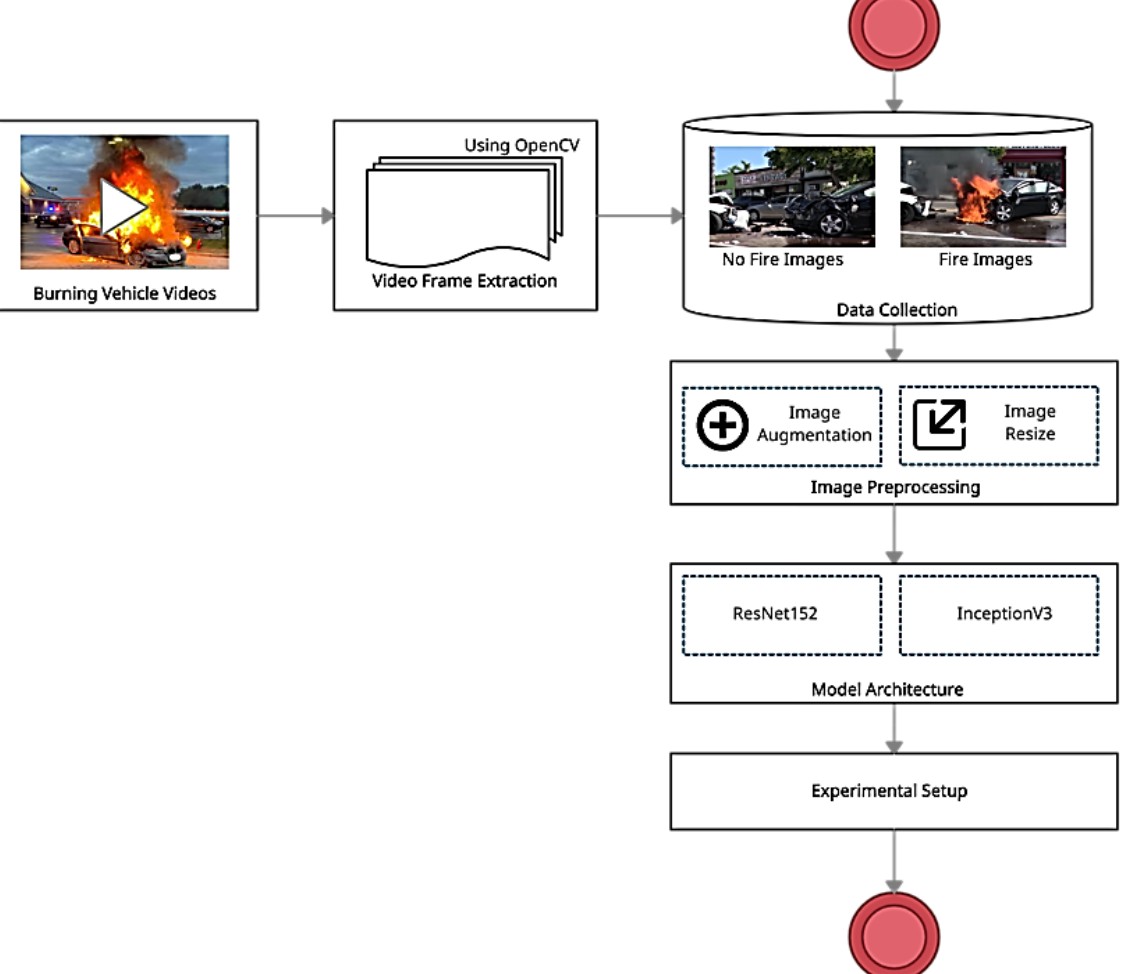

**Figure 5.** Phase 3 Flowchart for Detecting Post Collision Fires.

Accordingly, recent advancements in emerging technologies have permitted computer-vision based systems with the utmost ability to detect fires while utilizing convolutional neural networks, for which two regularly utilized image classification algorithms, the ResNet152 [23] and InceptionV3 [24] architectures, were implemented in the proposed work. This is to present an intelligent and automated vehicular postcollision fire detection approach that aims towards pertaining a high classification rate and a low false alarm rate at an expeditiously fast execution time to make an efficacious stride towards de-

tecting the occurrence of a fire after a traffic incident has taken place. The architectural workflow in Figure 5 shows how the postcollision vehicular fire detection model will be thoroughly implemented.

### 3.3.1. Data Collection

No suitable and befitting open-source dataset was found to help form a constructive representation of a real-world fire incident scheme. A custom collected dataset that procured two classes, fire, and no-fire, was created. Accordingly, burning vehicular incident images were collected and frames were extracted from YouTube videos. A disparate and differing combination of burning scenarios illustrated from different angles, atmospheric conditions, and illumination levels was meticulously considered. The fire class was comprised of 2750 images that encompassed any type of burning vehicular incident, including small fires and vehicle explosions. All were included within the dataset to capture all possible contingencies of postcollision fires. An additional 3200 images that contained no fire were collected. Figure 6 visually and pictorially displays a sample of the fire and no-fire images that were stored within the dataset.

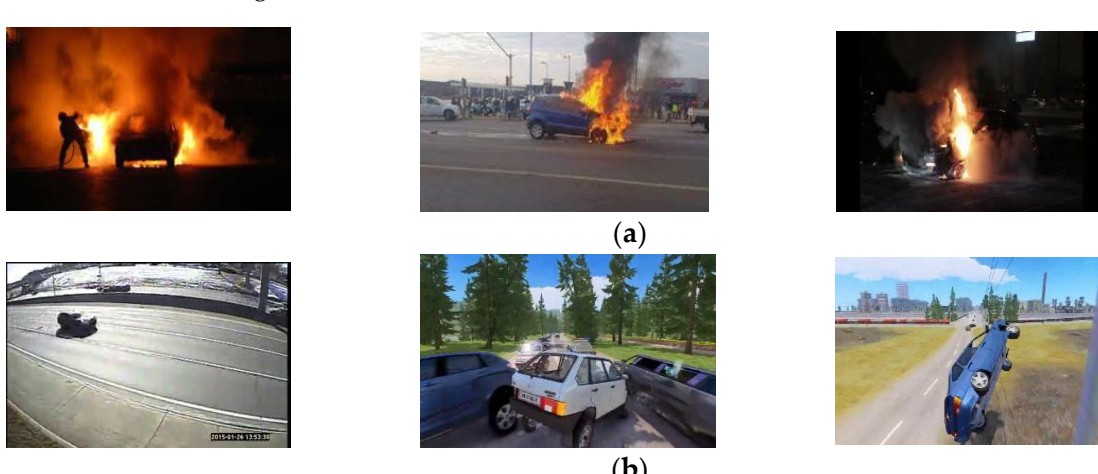

**Figure 6.** Sample of the Post Collision Dataset: (**a**) displays the fire images and (**b**) no-fire.

### 3.3.2. Image Preprocessing

The first and most prominent preprocessing step was to resize the images into a fixed width and height dimensionality. Downscaling the images, which lowers the dimensionality of the images, is the procedural process that was implemented, as it is a preeminent factor leading to a more efficacious model by relatively minimizing the model's overall computational complexity, training time, and memory consumption. Table 1 displays the image resize parameters for the two algorithms that were trained and tested for the vehicular postcollision fire detection model.

**Table 1.** Image Resize Parameters for ResNet152 and InceptionV3.

| Network | Pixel Dimensions |
| --- | --- |
| InceptionV3 | $299 \times 299$ |
| ResNet 152 | $256 \times 256$ |

Subsequently, to strategically increase the diversity and generalizability of the data available for training the model and to relatively decrease the probability of overfitting, an amalgam of image transformations was applied. The set of customarily utilized conversion transformation techniques that have been applied for the ResNet152 and InceptionV3 models include random resized crop, center crop, random horizontal flip, and normalization transformations along with converting the images into tensors. To add more images with

differing illumination levels, the brightness image level augmentation was implemented to the images allotted within the dataset by darkening or lightening the image by 25%, as this technique will aid the model towards being more resilient to abrupt changes that could potentially occur due to alterations and adjustments that are due to the abundant settings of CCTV cameras.

### 3.3.3. Experimental Setup

To constructively carry out the postcollision vehicle fire detection model, the set of images in the dataset were directly and randomly partitioned and segregated into three distinctive sets: the training, validation, and test sets. Where 80% of the images were utilized for beneficially training the model, 10% was allotted for validating the data and the remaining 10% was leveraged for testing the model to attain an unbiased viewpoint on the model's performance. Before training, the criterion variable that represented the loss function to assess and evaluate the effectiveness of the model's ability to model the given set of image instances by comparing the predicted class label to the actual label at a single point of time was specified. Accordingly, the cross-entropy loss function that has seemingly achieved state-of-the-art results in a wide spectrum of image classification tasks was used for both deep learning frameworks [25].

While training the deep learning model, the weights and learning rate after each epoch had to be adjusted and altered by a mathematical function to minimize overall loss and enhance the model's performance. This function is known as an optimizer, and its prominent intent is to bind the loss function and model parameters together by reforming the model in response to the output of the loss function [26].Consequently, the Adaptive Moment Estimation optimizer, also known as the Adam optimizer, was used, as this optimizer works extremely well for relatively large datasets and requires a minimal amount of storage memory space, which leads towards establishing a computationally efficient model that is easy to undertake and implement [26].

During training, once the model's performance stops further increasing or refining on the basis of the validation dataset, the model is immediately triggered to stop the training process [27]. This procedure is known as "early stopping," which is a neural network regularization technique that plays a role in preventing the model from being overfitted. The model at the time that the training is stopped can then be leveraged, as using this approach aids towards procuring acceptable generalization performance. Hence, the training cycle was terminated as soon as the validation loss increased when compared to the validation loss preliminary to the previous training epoch.

### 3.4. Phase 4: Email Alert System

Alerts are traditionally sent out to inform a particular third party that something significant has occurred or is likely to occur, as this systematic alert provides extra entailed information that can constructively indicate what protective actions should be taken. Nonetheless, the proposed work incorporates an automated alert system based on email notifications as by sending an alert, the needed emergency services can immediately dispatch the necessary medical assistance and notify authorities of the accident as quickly as possible to reduce and minimize the response time of medical help. When the CCTV traffic camera detects an accident, the alert system will automatically activate and go into effective action. An email will be sent to certain centers according to the severity level assigned to each accident as shown in Table 2.

The alert system was implemented using smtplib, which is a simple mail protocol that is accountable for sending and routing messages between servers, allowing automatic sending of emails while using the Python programming language. To guarantee the security of the servers within the system, the smtplib.SMTP_SSL() function was used. This automated alert system enables the General Department of Traffic and other decision-makers to see salient and detailed information about the accident in terms of its location and occurrence time. Hence, while using the Openpyxl Python library, all detected moderate and severe

accidents are saved as an Excel sheet report to maximize the value of the system. By obtaining this file, decision-makers can gain a deeper understanding of the nature and location of future incidents, and perhaps develop ideas about how to improve the department's infrastructure. This would also enable the department to collect and analyze accident information automatically and efficiently in an efficient and nonintimidating manner. Figure 7 illustrates the overall alert system's workflow that was deliberately undertaken.

**Table 2.** Email Alert System's Receiver and Email Content Description.

| Accident's Severity Level | The Center Receiving the Email | The Content of the Email |
|---|---|---|
| **Severe** | Medical Center | Severe Collision Occurred.<br>Accident Location:<br>(The IP address of the CCTV camera)<br>Incident time. |
| **Fire** | Fire Station | Fire Occurred.<br>Accident Location:<br>(The IP address of the CCTV camera)<br>Incident time. |

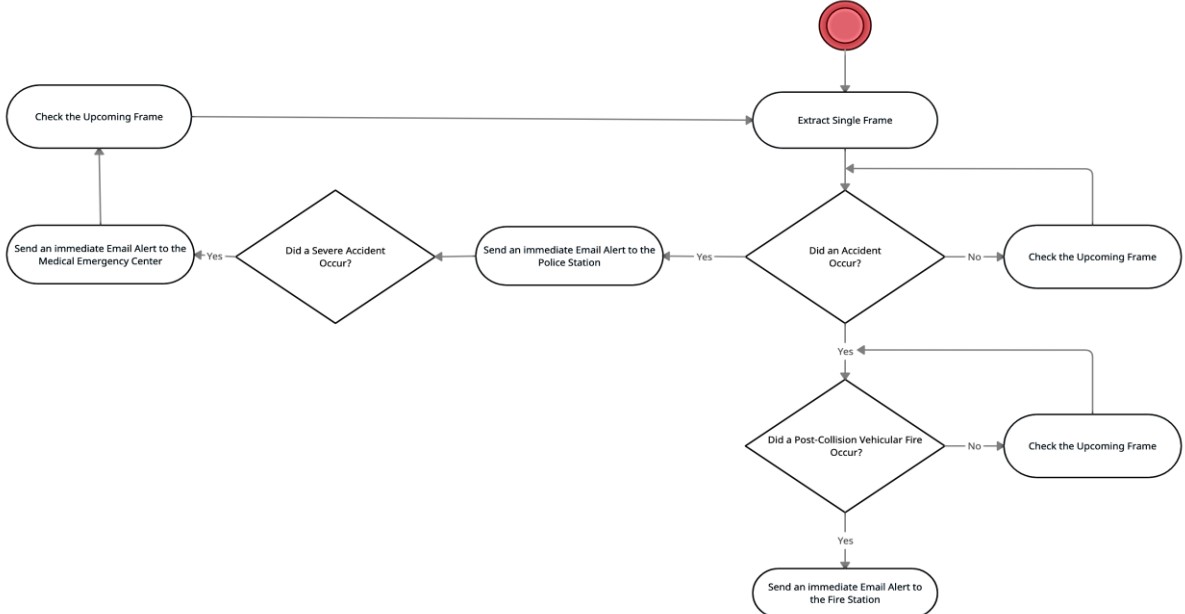

**Figure 7.** Email Alert System Workflow Flowchart.

*3.5. Phase 5: Model Integration*

The model integration phase, which is the final phase of the proposed system's pipeline, offers the finalized framework architecture that integrates the three premier object detection and image classification models along with the automated email alert system into a singular unified system by embedding it into a GUI. The QT Designer program was used to construct the GUI that covers all aspects of the proposed system while utilizing the Python programming language. Consequently, the computation time required for running a captured or live traffic surveillance video stream on three computer vision-based models remains a prominent limitational drawback and bottleneck, especially when the size of the tested video footage is relatively large, making the overall prediction and evaluation execution time slower. Accordingly, to obliterate and eliminate such consequences from occurring and to quickly speed up the model's overall inference time, a concurrent thread-based parallelization technique was utilized.

The multithreading procedure aids towards maximizing overall CPU or GPU optimization, as it will permit multi-core parallelism by possibly leveraging all available CPU or GPU cores. Three slave threads, including the GUI, video, and model threads, were formed using the QThread approach that is offered by QT to concurrently execute the tasks that will be assigned to each thread. The integration aspects' overall workflow architecture is displayed in Figure 8. Accordingly, the three established computer vision models were first loaded, where the video thread was prominently accountable for running the live or captured CCTV traffic video stream and extracting frames that will be sequentially and methodically passed to the second slave thread, known as the model thread. The model thread is the most salient and overriding slave thread, as the execution of the models will be constructively executed by this thread, which sanctions the detection vehicles, the classification of accidents, and the detection of postcollision vehicular fires. The frame counter variable was originally initialized to a negated value to keep track of the number of frames being passed to the slave thread.

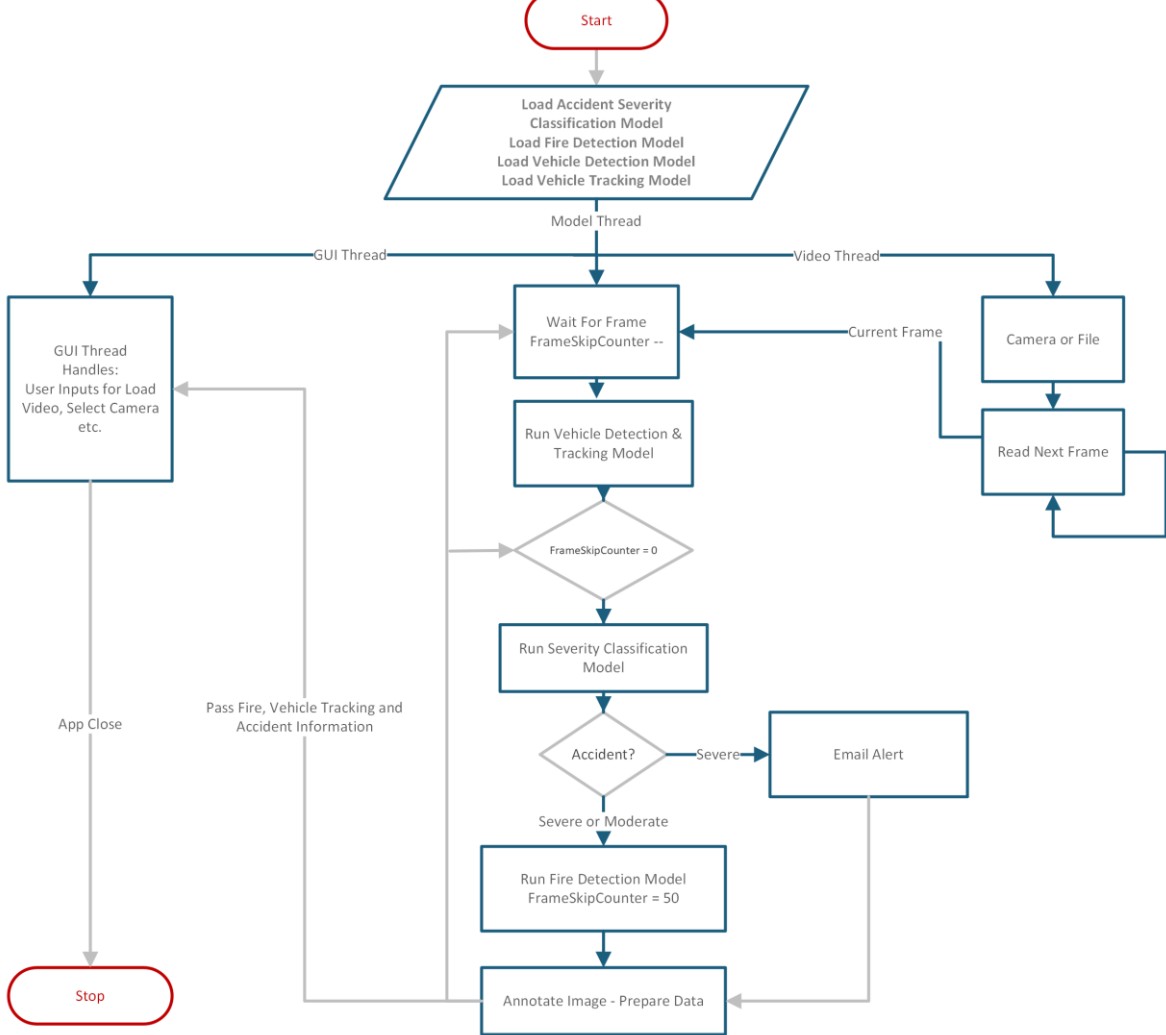

**Figure 8.** Proposed Systems Model Integration Flowchart.

Thereafter, the vehicle detection and tracking model was called to undergo the detection and tracking aspect of the proposed system, and then the counter was set to 0. The detected frame is then passed to the severity classification model to classify the extremity level of the accident if such an unforeseen event occurs. Once an accident, whether severe or moderate, has taken place in an appending series of video frames, all the subsequent video frames will be fed into the fire postcollision detection model to autonomously scru-

tinize and assess the detection of fires in such accident-based scenarios by sending an email alert to the competent and specified authorities. Specifically, if a severe accident occurs, an email alert will be sent to the nearest medical center for undertaking the needed medical measures. All the entailed information that was retrieved from the three models in the model thread is passed into the GUI thread to be graphically portrayed where this procedure is iteratively undertaken for each extracted frame, and the three established slave threads are concurrently running and executing in a parallel manner model to speed up and further ameliorate the overall inference speed of the constructed AI system.

## 4. Results

To determine the most optimally strategic and functional solution to the proposed AI-integrated traffic surveillance system, there is an imperative and rudimentary obligation to undergo an experimental iterative test cycle. This approach will untangle the multiple puzzling pieces of which algorithm is suitable to be permitted and required to undertake the specified course of action. Consequently, multiple experimental test runs were implemented while tweaking and adjusting a defined set of hyper-parameter values, including the number of batches, epochs, and iterations in each experiment. Adjusting the hyper-parameters is in fact a cardinal phenomenon in the computer vision and deep learning field, as they are simply knobs that can aid towards fine-tuning the model before pursuing the training and testing procedure to constructively control the overall behavioral analysis of the algorithm. The value of such hyper-parameter variables can have a significant and notable impact on the model's training as it plays a fundamental role in impacting the model's training time, neural network infrastructure, the model's convergence, and the model's overall performance accuracy. Accordingly, Table 3 displays the best experimental test runs that have been obtained with each of the three object detection algorithms while implementing the vehicle detection and tracking model. The YOLOv5 has proven to be the best-performing algorithm as shown in Table 3. Besides attaining a mAP of 99.2%, the reason this model was selected is the fact that it reached the highest precision rate of 98.4%, which is crucial when it comes to detecting vehicles and avoiding misclassifying vehicles in a moving traffic environment.

**Table 3.** Vehicle Detection and Tracking Most Optimal Experimental Results.

| Algorithm | Epoch/Iteration | Batch Size | mAP | Precision | Recall | F1-Score |
|---|---|---|---|---|---|---|
| YOLOv5 | 900 | 16 | 99.2% | 98.4% | 97.5% | - |
| YOLOR | 200 | 32 | 99.4% | 90.3% | 98.6% | - |
| YOLOv4-Tiny | 50,000 Iterations | - | 91.71% | 93.1% | 91% | 90.5% |

Like the vehicle detection and tracking model, for the traffic accident and severity classification model, three differing object detection algorithms have been tested and further evaluated to scrutinize their overall performance for comparison purposes to determine the algorithm that aims towards eliminating the delay between the traffic accident's occurrence and the first responder dispatch. Hence, Table 4 portrays in tabular form the most optimal results obtained in experimental test runs of each algorithm while implementing the accident severity classification model. It was found that the YOLOv5 algorithm was the most efficient algorithm, as it surpassed and outmatched the remaining algorithms and achieved the highest level of accuracy in detecting and classifying accidents' severity levels with a mAP of 83.3% and a precision rate of 84.01%.

For the postcollision vehicle fire detection model, a deep learning approach was integrated into the model's proposed framework to assess and further retrieve the approach that leads towards effectively and accurately detecting the occurrence of a fire after an incident is taken place. The approach utilizes an image classification technique that simply provides an algorithmic image analysis of the dataset by categorizing the image to fall within a particular class label. The ResNet152 and Inceptionv3 CNN architectures were

deployed for this task to effectively reduce and minimize the losses by autonomously alarming the required third party to undertake the entailed set of actions. Table 5 displays the most optimal results that have been obtained with each algorithm while specifying the epoch and batch size hyperparameters that were used throughout the process.

**Table 4.** Severity Classification Most Optimal Experimental Results.

| Algorithm | Epoch/Iteration | Batch Size | mAP | Precision | Recall |
|---|---|---|---|---|---|
| YOLOv5 | 1150 | 16 | 83.3% | 84.01% | 76.7% |
| YOLOR | 100 | 16 | 87.2% | 73.9% | 86.2% |
| Faster R-CNN | 30,000 Iterations | 16 | 79.8% | - | 68.6% |

**Table 5.** Postcollision Vehicular Fires Most Optimal Experimental Results.

| Algorithm | Epoch/Iteration | Batch Size | Accuracy/mAP | Precision | Recall |
|---|---|---|---|---|---|
| Resnet152 | 100 | 16 | 98.955% | 98.72% | 98.905% |
| InceptionV3 | 200 | 64 | 97.563% | 98.178% | 96.768% |

In this model, the ResNet152 neural network structure overtopped and outmatched the Inceptionv3 algorithm in terms of the four customarily utilized evaluation metrics that were used and are shown in Table 5. The epoch and batch size hyper-parameters that attained such a prodigious performance rate leveraged a batch size of 16 and an epoch value of 100, as this set of hyperparameter values can constructively and without a doubt detect the occurrence of fires in a real-time traffic-based environment.

Figure 9 pictorially and diagrammatically illustrates the learning curve representation that functionally compares the training and validation procedures in terms of accuracy and loss for the ResNet152 image classification-based algorithm. Furthermore, the learning curve acts as a diagnostic tool that attains the ability to further scrutinize the behavior of the model by assessing whether a good fit was achieved or not. As portrayed in Figure 9, a good fit was successfully achieved, as the training and validation losses tended to relatively decrease over time to the point that they attained stability as shown by the minimal margin between their computed loss values.

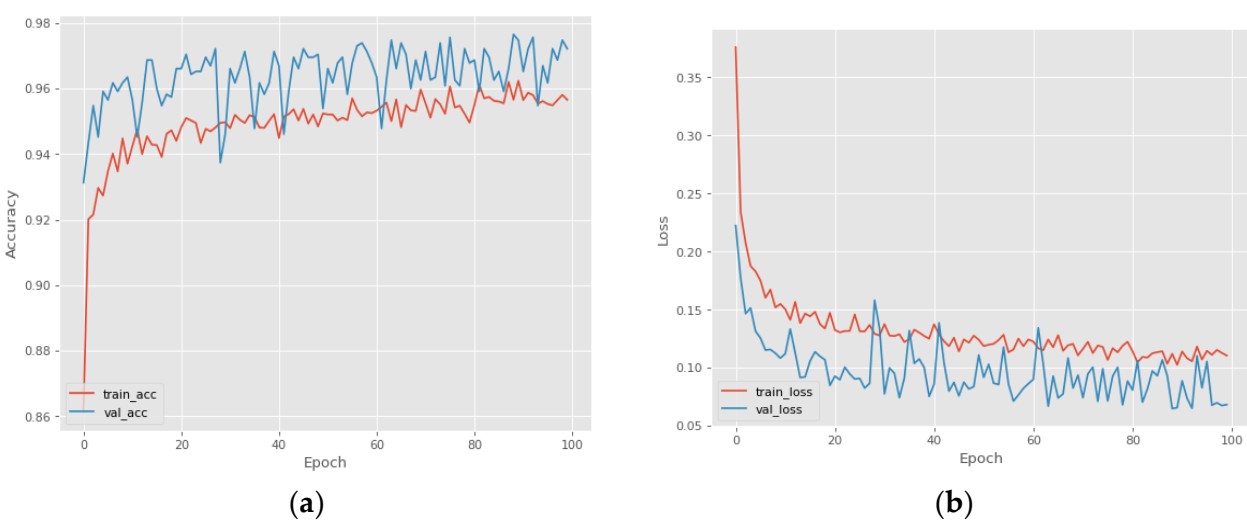

**Figure 9.** ResNet152 Learning Curves:(**a**) Accuracy, (**b**) Loss.

## 5. Discussion

Study findings and previous works indicate that traffic accidents are one of the leading causes of death and disability in Saudi Arabia, which means a better and more automated AI-based system for traffic safety is needed to make the country based on smart cities. The computer vision-based framework that was introduced in this proposed work resolves this societal and transportation-based problem by increasing the overall efficiency of transportation systems and by speeding up the time in which emergency or fire services dispatch first responders to the incident's victims to make a noteworthy difference in changing the impending sequence of events for the victims' lives. The proposed multifunctional framework is composed of an innovative and an unconventional selection of AI-based models including the vehicle detection and tracking model, the accident severity classification model, the postcollision fire detection model, and an automated email alert system that has been integrated using parallel computation techniques. Until now, no scientific research has exactly tackled and conducted this AI-based system; however, similar frameworks have been conducted and will be exhaustively compared in this section.

According to [28], a similar framework was implemented that combines a traffic monitoring model, a collision detection model, and a fire detection model along with an alert system. The framework in [28] incorporates some of the models our proposed system utilizes and detects accidents on the basis of using image frames that have been extracted from videos. As opposed to [28], the proposed work in this study incorporates a method for detecting accidents by further classifying its severity level as either moderate or severe. Additionally, the fire detection model used in [6] does not apply specifically to postcollision vehicular fires, which can come into play following an accident. As a result of the carried-out experiments for the proposed framework, the YOLOv5+DeepSORT algorithm that was used in the vehicle detection and tracking model attained a mAP of 99.2%. As compared to the state-of-the-art-algorithms that were employed in previous studies [29–32], the proposed algorithm attained the highest performance, followed by [29], as it achieved a mAP of 94.78% while using the YOLOv2 object detector as shown in Figure 10.

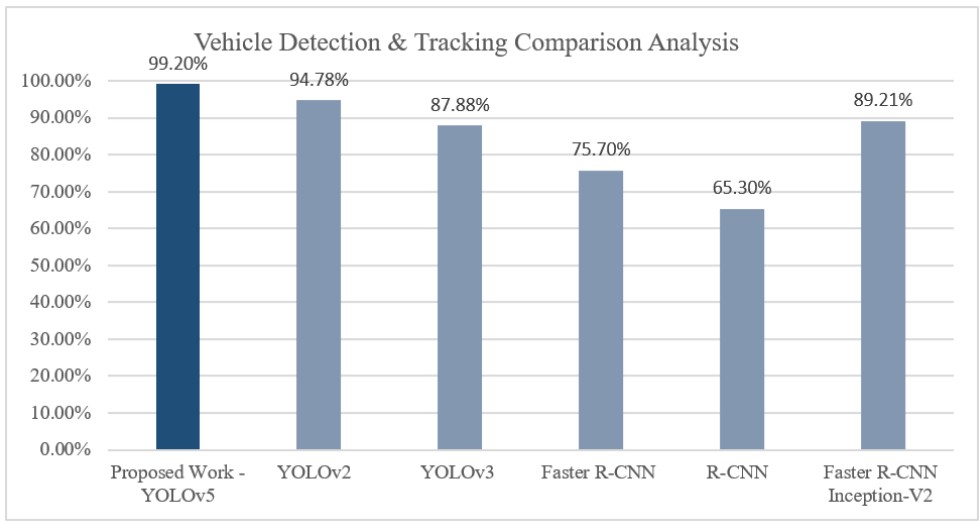

**Figure 10.** Vehicle Detection and Tracking Comparison Analysis Bar Chart.

The proposed approach for accident detection and severity classification applies the YOLOv5 object detector to an image-based dataset to determine the accident severity level, and it attained a mAP of 83.3%. As compared to different approaches used to assess accident severity, the researchers in [33] obtained an accuracy of 96% with the use of the SVM algorithm, which effectively classifies the vehicle images into either damaged or undamaged vehicles after an incident occurs, versus our proposed model, which detects accidents but further classifies their severity levels. This proposed work proposes an

innovative automated approach for classifying accidents' severities based on accident images, which is an approach that has not been investigated in previous studies.

Similarly, the postcollision vehicular fire detection model is novel, as no previous work has utilized a dataset that is solely based on burning automobiles. With an accuracy of 98.95%, the ResNet152 algorithm was the best-performing algorithm used in the post-collision fire detection model. Prior research has mostly focused on customarily utilized models that detect the occurrence of a fire under any unforeseen circumstances, including building fires and wildfires. The researchers in [34] obtained a significantly high mAP of 99.62% by using the YOLOV3 algorithm to identify generalized occurrences of fire. The same study [34] also showed that Faster-RCNN had a mAP of 99.43% in detecting fire and smoke. Accordingly, the results showed higher accuracy rates for detecting fire incidents in comparison to our novel model; however, the fire event that focuses on the detection of fires after a vehicular collision occurs was not considered.

## 6. Conclusions

Many traffic accident deaths are attributed to delays in the emergency response. Accordingly, to improve traffic movement and increase traffic safety, the proposed AI system was designed using AI techniques including deep learning and computer vision to detect and track vehicles, classify the accidents' severity levels, and detect postcollision vehicular fires to send a notification alert to the nearest hospital or fire station if needed to take the appropriate measures. The first model was constructed to detect and track vehicles in a real-time manner by deploying the YOLOv5 algorithm along with the DeepSORT tracker, and it attained a mAP rate of 99.2%. The second model, which is the core of the system, was established using the YOLOv5 algorithm to efficiently detect and classify the accident's severity level, and it attained a mAP of 83.3%. The third model, which aims towards classifying fire occurrences after an incident has taken place, utilized a transfer learning algorithm known as ResNet152, and it has successfully achieved an accuracy rate of 98.955%.

The three models were constructed and implemented using Google Colab's Virtual Machine and were trained while leveraging the NVIDIA DGX Personal AI Supercomputer to speed up the training process. Consequently, all three models along with the alert system were parallel integrated into a GUI as it could potentially be embedded within CCTV traffic surveillance cameras in the future to demonstrate the effectiveness of capturing the occurrence of an accident and sending an immediate notification alert to the right dispatch. As part of future work, developing a license plate localization model will be considered to further expand and ameliorate the system's capabilities to retrieve the license plate numbers of the vehicles involved in the accident. Additionally, we aim to improve the performance of the accident detection and severity classification model by expanding the dataset and increasing the number of epochs during training. Finally, this system was designed to contribute to the Kingdom's Vision 2030, particularly in the smart cities objective. Therefore, this system should be deployed and experimented on real CCTV traffic surveillance cameras in regions where the traffic incident rate is high within Saudi Arabia to reduce traffic accidents and minimize their ruinous repercussions. In future, we intend to investigate various other Machine Learning and Deep Learning models [35] to further fine tune an enhanced performance of the proposed model.

**Author Contributions:** Conceptualization, R.S.; methodology, R.A., R.S., J.A., S.A. and B.A.A.S.; software, R.A., R.S. and J.A. validation, R.A., R.S., J.A., S.A. and B.A.A.S.; formal analysis, S.A., B.A.A.S., R.A., R.S. and J.A. investigation, J.A., R.S., S.A., B.A.A.S. and R.A.; resources, R.A., R.S., J.A., S.A.,B.A.A.S. and M.I.B.A.; data curation, R.A., R.S., S.A. and J.A.; writing—original draft preparation, R.A., R.S., J.A., S.A., B.A.A.S. and M.I.B.A.; writing—review and editing, M.I.B.A., R.Z., M.S.A., A.R. and G.K.; visualization, R.S., J.A., R.A., S.A. and B.A.A.S.; supervision, M.I.B.A., R.Z., M.S.A., A.R. and G.K.; project administration, M.I.B.A. All authors have read and agreed to the published version of the manuscript.

**Funding:** This research received no external funding.

**Informed Consent Statement:** Not Applicable.

**Data Availability Statement:** Not Applicable.

**Conflicts of Interest:** The authors declare no conflict of interest.

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
