# Peer review of "A Real-Time Computer Vision Based Approach to Detection and Classification of Traffic Incidents"

_2504-2289, doi:10.3390/bdcc7010022_

Round 1
Reviewer 1 Report
The topic describes the computer vision based on method for traffic incidents. This manuscript looks fine at some point. I recomment to accept after the following issues are solved.
1, Any novelty of the methods? what is the purpose to choose this topic as a research?
2, There are still many method for vision learning, the author can cite more papers to support the background, such as Micromachines. 2022, 13, 332.
3, The text in some figures looks small, please make them in a proper size.
4, What about the copyright of Figure 3?
5, What the reason to make the results difference in different method?
Author Response
Response to Reviewer’s Comments
Authors are grateful for the valuable comments/suggestions provided by the reviewers. We have tried to closely follow up with them.
----------------------------------------------
Reviewer 1:
The topic describes the computer vision based on method for traffic incidents. This manuscript looks fine at some point. I recommend accepting after the following issues are solved.
1, Any novelty of the methods? what is the purpose to choose this topic as research?
Response: In the KSA region, traffic accidents are mainly evident on the national highways due to several reasons such as sandstorms, camels crossing etc. Due to the huge geographical road expansions, physical check posts and patrolling are not applicable. Hence, there is a dire need of such systems to get the timely alert and response. State of the art dataset from the middle eastern region has been investigated in this regard.
2, There are still many methods for vision learning, the author can cite more papers to support the background, such as Micromachines. 2022, 13, 332.
Response: The said study has been cited as reference [29].
3, The text in some figures looks small, please make them in a proper size.
Response: The images has been enhanced to increase the text readability.
4, What about the copyright of Figure 3?
Response: The images belong to an open-source dataset and does not bear any copyright formality.
5, What the reason to make the results difference in different method?
Response: To have holistic and diverse analyses, various models have been investigated over the same dataset. That resulted in the diversified outcomes of various levels.
Reviewer 2 Report
Below are more specific comments:
1. Line 161. Lots of inappropriate self-citations. [24-35] The details of these work were not discussed and it’s not obvious why these cited works are relevant to the current manuscript.
2. Line 181. Is email alert an effective tool as emails are often used for non-urgent communications? Will the first responder systems monitor emails and dispatch accordingly?
3. Line 226. How was this custom dataset annotated?
4. Line 330. CNN-based algorithms was a crucial step the was implemented -> CNN-based algorithms was a crucial step which was implemented
5. Some sentences are too lengthy (e.g. Line 344-351) and should be shortened. For example, “the next indispensable and integral task of the proposed model is to validly keep track of each of the detected vehicular objects.” These words can be removed: indispensable, integral, validly.
6. Fonts in the figures of this manuscript should be enlarged for better readability.
7. Will the authors make the datasets used in this study public?
Author Response
Response to Reviewer’s Comments
Authors are grateful for the valuable comments/suggestions provided by the reviewers. We have tried to closely follow up with them.
----------------------------------------------
Reviewer 2:
Below are more specific comments:
- Line 161. Lots of inappropriate self-citations. [24-35] The details of these work were not discussed and it’s not obvious why these cited works are relevant to the current manuscript.
- Line 181. Is email alert an effective tool as emails is often used for non-urgent communications? Will the first responder systems monitor emails and dispatch accordingly?
Response: In the current system prototype, email module was focused as an alert method. However, in the extension, we will use other types of alerts as well such as push notification from the APP and GSM based notification etc. Point is taken.
- Line 226. How was this custom dataset annotated?
Response: The open-source dataset used for the experiment was pre-annotated.
- Line 330. CNN-based algorithms were a crucial step the was implemented -> CNN-based algorithms were a crucial step which was implemented.
Response: The recommendation has been incorporated with thanks.
- Some sentences are too lengthy (e.g., Line 344-351) and should be shortened. For example, “the next indispensable and integral task of the proposed model is to validly keep track of each of the detected vehicular objects.” These words can be removed: indispensable, integral, validly.
Response: The recommendation has been incorporated with thanks.
- Fonts in the figures of this manuscript should be enlarged for better readability.
Response: The recommendation has been incorporated with thanks.
- Will the authors make the datasets used in this study public?
Response: The dataset is already available publicly.
Reviewer 3 Report
In order to increase the needed demand for road traffic security and safety, this paper presents a real-time incident detection and alert system developed using computer vision (YOLOv5 and ResNet152 algorithms). The authors have shown that the proposed system produces high accuracy rate. While the work is very interesting and worth to be investigated, I have some minor comments.
1) This paper is very interesting and challenging. Moreover, this paper is well-organized and well-written.
2) The authors need to clearly state the contribution of their work. It is advised that they list them in points form at the end of the introduction section.
3) A deeper and more thorough analysis of the related works needs to be provided. What are the limitations of the related works? What research gap is this work filling? This needs to be clearly discussed and stated.
4) Paper needs proofreading. There are multiple grammatical and typing errors. Please review the paper since these errors significantly impact the readability of the paper.
5) A clearer presentation of the dataset needs to be provided. For example, what is the total number of features? The data preprocessing steps need to be discussed better. Additionally, the authors must provide references to the open-source data used in this study.
6) The Result and Discussion section can be improved; it is suggested to strengthen this part with more details and justifications.
7) Quality of the figures is low. Increase the text font size and use the same font size for all the images. Thus, all the figures need to go through a quality check and must be improved.
8) lumped references should be removed.
Author Response
Response to Reviewer’s Comments
Authors are grateful for the valuable comments/suggestions provided by the reviewers. We have tried to closely follow up with them.
----------------------------------------------
Reviewer 3:
To increase the needed demand for road traffic security and safety, this paper presents a real-time incident detection and alert system developed using computer vision (YOLOv5 and ResNet152 algorithms). The authors have shown that the proposed system produces high accuracy rate. While the work is very interesting and worth to be investigated, I have some minor comments.
1) This paper is very interesting and challenging. Moreover, this paper is well-organized and well-written.
Response: Gratitude.
2) The authors need to clearly state the contribution of their work. It is advised that they list them in points form at the end of the introduction section.
Response: Contribution list added at the end of introduction section.
3) A deeper and more thorough analysis of the related works needs to be provided. What are the limitations of the related works? What research gap is this work filling? This needs to be clearly discussed and stated.
Response: Provided at the end of introduction section.
4) Paper needs proofreading. There are multiple grammatical and typing errors. Please review the paper since these errors significantly impact the readability of the paper.
Response: Paper has been revised for typo grammatical errors.
5) A clearer presentation of the dataset needs to be provided. For example, what is the total number of features? The data preprocessing steps need to be discussed better. Additionally, the authors must provide references to the open-source data used in this study.
Response: The citations to the datasets has been added.
6) The Result and Discussion section can be improved; it is suggested to strengthen this part with more details and justifications.
Response: The said section has been updated.
7) Quality of the figures is low. Increase the text font size and use the same font size for all the images. Thus, all the figures need to go through a quality check and must be improved.
Response: Quality of the images has been improved and text is readable.
8) lumped references should be removed.
Response: The said references are removed/updated.
Reviewer 4 Report
This paper tried to propose a CV-based method for real-time traffic incident detection and alert system.
1. The novelty seems pretty limited since the model is actually incremental. This work does not provide a new or improved model. The contribution of this paper presents simply a workflow that has been widely investigated in the past decade.
2. The datasets in experiments are relatively small.
3. How does the proposed model handle outliers or noisy data? It is very important to evaluate the robustness of the model for traffic predictions.
4. The authors may consider adding more experimental results by comparing the proposed model to some SOTA methods.
Author Response
Response to Reviewer’s Comments
Authors are grateful for the valuable comments/suggestions provided by the reviewers. We have tried to closely follow up with them.
----------------------------------------------
Reviewer 4:
This paper tried to propose a CV-based method for real-time traffic incident detection and alert system.
- The novelty seems pretty limited since the model is actually incremental. This work does not provide a new or improved model. The contribution of this paper presents simply a workflow that has been widely investigated in the past decade.
Response: In terms of novelty, the article investigates various algorithms over various dataset for vehicle, accident, and fire detection separately. In KSA, so far no such study has been conducted.
- The datasets in experiments are relatively small.
Response: Three different datasets has been investigated segregated by the vehicle, accident, and fire detection. Separate set of experiments have been conducted under different algorithms for each type of dataset.
- How does the proposed model handle outliers or noisy data? It is very important to evaluate the robustness of the model for traffic predictions.
Response: In the dataset preprocessing various techniques for image denoising, scaling and transformation have been used. As explained in the relevant section.
- The authors may consider adding more experimental results by comparing the proposed model to some SOTA methods.
Response: Since the dataset is from different sources, augmented and annotated by the authors. In terms of comparison with the SOTA, it may not be fair comparison in the current setup.
Round 2
Reviewer 1 Report
The author answer all my issues. I recomment to accept in the current form